# Reconstitution of Caveolin-1 into Artificial Lipid Membrane: Characterization by Transmission Electron Microscopy and Solid-State Nuclear Magnetic Resonance

**DOI:** 10.3390/molecules26206201

**Published:** 2021-10-14

**Authors:** Yanli Zhang, Xinyan Zhang, Wenru Kong, Shuqi Wang

**Affiliations:** 1Department of Pharmacy, Qilu Hospital of Shandong University, Cheeloo College of Medicine, Jinan 250012, China; yanlizhangqlyy@163.com; 2Department of Pharmaceutical Sciences, University of Nebraska Medical Center, Omaha, NE 68198, USA; xinyan_zhang@hms.harvard.edu; 3School of Pharmaceutical Sciences, Shandong University, Jinan 250012, China; qfkongwenru@163.com

**Keywords:** caveolin-1, lipid membranes, membrane curvature, ssNMR, α-helical, caveolin scaffolding domain

## Abstract

Caveolin-1 (CAV1), a membrane protein that is necessary for the formation and maintenance of caveolae, is a promising drug target for the therapy of various diseases, such as cancer, diabetes, and liver fibrosis. The biology and pathology of caveolae have been widely investigated; however, very little information about the structural features of full-length CAV1 is available, as well as its biophysical role in reshaping the cellular membrane. Here, we established a method, with high reliability and reproducibility, for the expression and purification of CAV1. Amyloid-like properties of CAV1 and its *C*-terminal peptide CAV1(168-178) suggest a structural basis for the short linear CAV1 assemblies that have been recently observed in caveolin polyhedral cages in *Escherichia coli (E. coli).* Reconstitution of CAV1 into artificial lipid membranes induces a caveolae-like membrane curvature. Structural characterization of CAV1 in the membrane by solid-state nuclear magnetic resonance (ssNMR) indicate that it is largely α-helical, with very little β-sheet content. Its scaffolding domain adopts a α-helical structure as identified by chemical shift analysis of threonine (Thr). Taken together, an in vitro model was developed for the CAV1 structural study, which will further provide meaningful evidences for the design and screening of bioactive compounds targeting CAV1.

## 1. Introduction

The cellular membrane performs a wide range of essential biological functions, some of which are extremely complex and often require equally elaborated rearrangements of the membrane. One specialized membrane domain in mammals, named caveolae, is very easily recognized by its typical omega-shaped, intra-cellular membrane invagination when observed using electron microscopy [1,2,3]. Although discovered in the early 1950s, the caveolae function has only been explored in the past two decades [4]. Today we know that caveolae are implicated in endocytosis, mechanosensing, signaling, and lipid trafficking and metabolism [5,6,7]. Consistent with their multiple roles in cell biology, caveolae are abundant in many cell types and are involved in numerous diseases ranging from viral and bacterial infections, lipodystrophy, muscular diseases, cardiac disease, and cancer to neurodegenerative diseases [8,9]. Caveolae are therefore extremely important biomedical substructures of the mammalian cellular membrane. The flux of information about caveolae was triggered by the discovery of a protein marker for caveolae: the caveolin (CAV) membrane protein [10,11]. Three proteins are found in the caveolin family. CAV1 and CAV2 are expressed in non-muscle cells. Skeletal and cardiac muscle cells only express CAV3, while smooth muscle expresses all three isoforms. CAV1 or CAV3 are found to be necessary for the formation of caveolae, depending on the tissue, whereas CAV2 is dispensable in vivo [12]. Caveolins not only regulate caveolae formation, but also are directly responsible for the relationships between caveolae and certain diseases. For example, caveolins has been directly implicated in breast, prostate, and lung cancers, and even as an essential modulator of chemoresistance. However, the role of caveolins in cancer remains controversial because it can function as both tumor suppressors and promoters and more studies are needed to elucidate these findings [13,14,15]. Involvement of caveolins in other diseases is however better understood, such as in diabetes, Alzheimer’s disease, lung injuries, and infections [16,17]. Caveolin membrane proteins therefore emerge as extremely multi-potent pharmacological targets.

Despite their great potentiality as drug targets, structural information of caveolins is rather limited. In current study, we will focus on the research of CAV1, which has two isoforms: a α-isoform CAV1(1–178) and an *N*-terminal truncated β-isoform CAV1(32–178) missing amino acid residues 1–31 of the α-isoform. These isoforms have different but overlapping subcellular distributions and may contribute to small morphological differences in the caveolae that they form [18]. In the case of α-isoform of CAV1, it contains four domains: an *N*-terminal region (residues 1 to 82), a scaffolding domain (residue 83 to 102), a hairpin-like intramembrane domain (residues 103 to 134), and a *C*-terminal domain (residues 135 to 178). It was demonstrated that CAV1 functions in the form of large aggregates: either CAV1 homo-oligomers, or hetero-oligomers involving both CAV1 and CAV2, or CAV1 and CAV3 [19,20].

Various biochemistry and biophysical techniques have been utilized to examine the structural features of CAV1 and its oligomerization mechanisms. Since full length CAV1 tends to form large aggregates, traditional methods, such as solution NMR and X-ray crystallography, are usually not applicable for its structural characterization. Therefore, several short CAV1 truncates have been prepared and studied by circular dichroism (CD) and solution NMR. The *N*-terminal region (residues 61 to 101,), and the *C*-terminal region are proved to be indispensable for CAV1 oligomerization. The oligomers are ~8S in size determined by velocity gradient centrifugation, and are estimated to contain 7 to 14 copies of CAV1. Single particle electron microscopy of recombinant CAV1 complexes has shown that they are toroidal structures about 15 nm in diameter composed of an outer ring, an inner ring, and a central protruding stalk [21,22,23]. The intramembrane part was found to adopt α-helical structures with a break at proline 110, which governs both its structure and solvent exposure [24,25]. Both α-helical and beta sheet structures have been reported for the scaffolding domain, and a series of coarse-grain molecular simulations reveal that its secondary structure plays an important role in membrane curvature [26,27,28]. Residues of 132 to 175, representing the C-terminus of CAV1, exhibited α-helical structures [29]. Until now, structural studies regarding full length CAV1 are lacking, except that a full-length, soluble, human CAV1 variant was prepared by protein engineering and its secondary structure is a mixture of alpha helices and beta strands as revealed by CD [30].

ssNMR have been developed for the structural study of solid materials and been successfully applied in the elaboration of conformation and dynamics of Aβ fibrils, cell wall, and influenza virus membrane protein M2 [31,32,33]. In the current study, a simple and well-defined in vitro system of synthetic lipids and recombinant CAV1 was established to demonstrate the effect of CAV1 on membrane morphology, as well as the structural features of CAV1. This task requires the reconstitution of recombinant CAV1, which can be obtained from bacteria according to the study above, into artificial membranes. Such a platform would allow precise control over the lipid composition that could incorporate other relevant proteins, and would provide a functional assay for caveolin as a membrane scaffolding agent and drug targets, facilitating high-resolution structural studies and screening of drug candidates. We found that CAV1 induces not only the formation of membrane curvature, but also caveolae-like membrane invaginations. Extensive aggregation possibly involving amyloid-like structures is observed, providing a structural basis for short, linear CAV1 oligomers that have been recently reported. According to the structural information obtained by CD and ssNMR, it was determined that CAV1 is a mixture of alpha helices and beta sheets and the scaffolding domain is helical. These findings will be useful for the design and evaluation of bioactive compounds targeting CAV1.

## 2. Results

### 2.1. Protein Production

CAV1 expression was found to be greatest in the BL21 (λDE3) strain with saturated cultures at 27 °C. CAV1 is always obtained as a high-molecular weight oligomer that is highly resistant to detergents, as previously observed by other researchers and confirmed below [34]. It is therefore very likely that detergents do not change the CAV1 oligomeric structure and then several detergents were screened to find a simple protocol that maximizes the yield of purified protein. The cells were chemically lysed using a chemical cocktail containing Triton X-100 to partially solubilize membranes, which were then completely solubilized with empigen BB. CAV1 purification was achieved firstly over a nickel resin using empigen BB and then over an anion exchange resin while also replacing empigen BB with octyl-β-glucoside (BOG), for further reconstitution. The resulting CAV1 preparation is very homogeneous as observed in the elution peak shown in Figure 1A, suggesting that the charge surface distribution of CAV1 oligomers is very uniform. About 6 mg of pure CAV1 per liter of culture was consistently obtained under these conditions.

### 2.2. Protein Oligomerization

Previous studies demonstrated that CAV1 exists either as a ~350 kDa oligomer (major species) or as a ~200 kDa oligomer (minor species) and that both species can be disaggregated into monomers by boiling in SDS prior to gel electrophoresis [35]. The CAV1 purity was analyzed by SDS-PAGE and its histidine tag was confirmed by western blot using an anti-penta histidine antibody, as shown in Figure 1B. CAV1 appears to be pure and runs at the correct apparent molecular weight as a monomer (22–23 kDa); however, a significant part of the sample is found as large oligomers—with or without boiling in SDS—in this large-scale preparation. A more systematic investigation of the CAV1 oligomerization was then performed to elucidate this discrepancy.

CAV1 solubilization to a monomeric form by various detergents was not possible as monitored by size exclusion chromatography using a Superdex 200 Increase 10/300 GL column (GE Healthcare): all the protein were eluted in the void volume. Several common organic solvents and mixtures were then tested to obtain monomeric CAV1 at the expense of protein unfolding. When being analyzed over a Superdex 75 10/300 GL column, the FMA mixture of 20% each of formic acid, methanol, and acetonitrile in water showed the highest potential to break down CAV1 aggregates. Then, the protein was precipitated in methanol/chloroform, redissolved in neat formic acid, diluted with FMA, and injected onto the Superdex 75 column in FMA. About 60% of the CAV1 protein was, under these very adverse conditions, consistently purified in a monomeric form (Figure 1C) as judged by comparing the elution profile to that of soluble proteins of known molecular weights.

Because of concerns related to the CAV1 oligomerization, MS was employed to unambiguously identify CAV1. The FMA-purified CAV1 was then analyzed by (i) MS revealing that the CAV1 sample is very pure and corresponds to the expected molecular weight minus a methionine residue (Figure 1D) and (ii) by *N*-terminal sequencing by Edman degradation and identified the SGGKY sequence of CAV1(2–6), further confirming that the *N*-terminal methionine was removed most likely during bacterial expression. Protein trypsination within the excised SDS-PAGE band with In-Gel Tryptic Digestion Kit resulted in the MS identification of the peptides CAV1 (6–19), (48–57), and (87–96) (Figure 1E).

The 11-residue peptide CAV1(168-178) was found to exclusively form long amyloid fibrils as observed by TEM (Figure 1F), while aggregates formed by CAV1 are much shorter compared with the fibrils formed by CAV1 C-terminus (Figure 1G). This might be attributed to the steric constraints imposed by the CAV1 itself or by further interactions with C- and *N*-terminus. However, as revealed by standard ThT fluorescence assay (Figure 1H), both aggregates of CAV1(168–178) and CAV1 binds with ThT and causes the fluorescence enhancement, suggesting the existence of amyloid-like structures [36]. No further attempt was made here to prevent or disrupt CAV1 self-assembling as this may have a crucial biological role in caveolae formation. The following work therefore focused on characterizing as well as possible the CAV1 assemblies.

### 2.3. Membrane Reconstitution

CAV1 was firstly reconstituted into a mixture of DMPC and CHOL synthetic bilayers (molar ratio of 1:1) by detergent dialysis. BOG-solubilized lipids and CAV-1 were mixed together and incubated at 30 °C to allow the equilibration of the mixed micelles. The detergent was then extensively dialyzed out, resulting in the formation of CAV1-containing liposomes. Correct insertion of CAV1 into membranes competes with the generation of CAV1 soluble oligomers (which are also found in vivo) and non-specific aggregates. The former is removed by ultracentrifugation at 45,000× *g* (oligomers will remain in suspension), while the later will be removed via extrusion through 200 nm membrane filters (aggregates are too large and rigid to pass through). The protocol was optimized to minimize these competing pathways. Reconstitution with CAV1/total phospholipid weight ratios between 1:50 and 1:2 was tested, and ratios of 1:20 was used for TEM and 1:5 for NMR studies. CD was employed to detect changes in CAV1 secondary structure upon reconstitution. As shown in Figure 1I, the protein exhibits extensive helical content when solubilized in BOG detergent and a virtually identical spectrum upon their reconstitution into artificial membranes.

TEM was employed to systematically investigate the changes in membrane morphology induced by the incorporation of CAV1. The exact location of small membrane proteins cannot be inferred without specific labeling under these conditions. 5 nm Ni-NTA-Nanogold offers a good compromise between ease of use and labeling efficiency and was employed in this study [37]. It is noteworthy that this nanogold labeling strategy is limited to the protein population that fully exposes C-terminus plus the histidine tag outside of the liposome.

TEM of control DMPC/CHOL liposomes without protein (Figure 2A,B) confirms that the dialysis method produces heterogeneous uni- and multi-lamellar liposomes with diameters typically larger than 100 nm. The observed membranes are smooth, occasional defects are rarely observed (and are not consistent throughout the samples), and non-specific nanogold labeling is virtually absent. Isolated liposomes have spherical or elliptical shapes that are easily recognized. Most membrane deformations are due to physical contact between adjacent liposomes as they often cluster during absorption on the carbon film. Figure 2C,D show that TEM of control samples with CidA and LrgA proteins, which are also small integral membrane proteins involved in bacterial programmed cell death and biofilm formation [38]. Both proteins have been purified and characterized in our labs by the above methods and have been similarly reconstituted by dialysis into DMPC/CHOL membranes, except that *n*-dodecyl-N, *N*-dimethylamine-*N*-oxide (LDAO) was used instead of BOG. Unlike caveolins, these proteins are not expected to change the membrane curvature (it is hypothesized that they form pores in the membrane) and only form small oligomers that are readily disrupted by detergents under reducing conditions [39]. At low protein density, the CidA distribution on the membrane appears to be random and no protein aggregation is observed: the liposomes are sparingly, but uniformly, labeled with nanogold. At higher protein density, exemplified for LrgA, most liposomes are heavily, but still uniformly decorated with nanogold. Interestingly, LrgA is preferentially labeled with nanogold on the rim of the dry liposomes deposited on the carbon film. Nevertheless, the liposomes appear largely undisturbed by the incorporation of membrane proteins. Collectively, these controls show that the dialysis method for protein reconstitution is not intrinsically prone to induce protein aggregation or changes in membrane morphology and that any such effects must be due to specific protein–protein and protein–membrane interactions.

Reconstitution of CAV1 into DMPC/CHOL membranes is completely different: CAV1 self-assembles to cover distinct patches of the membrane, and causes budding out of the membrane domains. In vivo, CAV1 inserted into the inner leaflet of cellular membrane via the intramembrane part, with both the *C*- and *N*-terminus facing cytoplasm [40]. However, using the protocol described above, the CAV1 inserts into the synthetic lipid bilayer randomly. Formation of both evagination and invaginations would be reasonable. As shown in Figure 2E, a membrane structure resembling caveolae is exemplified, although it is a membrane ex-vagination unlike the invaginating caveolae. The diameters of these buds are between 20 and 80 nm (typical caveolae being 50 to 80 nm), which is in good agreement with the 30 and 120 nm diameter of the caveolae-like vesicles observed in bacteria. However, these buds are rarely seen in isolation as most of them cluster together, as depicted in the lower half of Figure 2F. These membrane clusters are so dense that it is difficult to discern individual structures, especially in the presence of the overwhelming nanogold particles. Nevertheless, buds smaller than 80 nm are always observed at the edges of the aggregates. Considering the oligomerization properties of CAV1 described above and the absence of stabilizers of the liposomes, large CAV1 self-assemblies and clusters of CAV1-induced membrane buds are to be expected under the conditions employed here. Although DMPC is a popular synthetic lipid for liposome formulation, DMPC/CHOL holds little biological relevance with respect to caveolae and other lipid formulations were tested. Thus, several other lipid mixtures were further employed for the reconstitution using the above protocol.

1,2-dihexadecanoyl-*sn*-glycero-3-phosphocholine (DPPC)/CHOL, a model of lipid rafts, has been implemented in molecular dynamics simulations of a caveolin peptide [41,42]. Similar to DMPC/CHOL, CAV1 also self-assembles over distinct membrane domains and exerts the same effects on DPPC/CHOL membranes. Figure 3A shows a 20 nm membrane bleb that occurs on the outer layer of a double-layer small liposome. Some of these buds seem to rupture when absorbed to and dried on the carbon film, resulting in the formation of a membrane discontinuity with adjacent nanogold-labeled material observed, as shown in Figure 3B,C for small and, respectively, large buds. Most of the CAV1 reconstituted into DPPC/CHOL also co-localizes within large membrane-protein aggregates (Figure 3D). In order to eliminate the possibility that the nanogold particles contribute to membrane deformation, the samples were subsequently imaged without nanogold labeling. As shown in Figure 3E,F, these clusters are indeed formed by the aggregation of many small vesicles with diameters smaller than 80 nm. Most of these vesicles appear to have an open end suggesting they are buds, rather than intact vesicles. Any additional mechanical forces exerted by the nanogold appear to have minimal impact on the membrane structure as studied here.

Since hexadecanoyl sphingomyelin (SM) is an essential component of caveolae, CAV1 was then reconstituted into the DPPC/SM/CHOL bilayer (Figure 4). Most of the control liposomes (without CAV1, Figure 4A) burst, adhere flat on the carbon grid, and appear to have two distinct domains under negative staining. These domains are very likely to be SM/CHOL-rich and -low domains. However, the phase separation is not easily observed in CAV1 containing liposomes, which remain intact and cluster extensively, suggesting the role of CAV1 in the redistribution of different lipids species in the membrane (Figure 4B). Reduced nanogold labeling (Figure 4C,D) reveals the same clustering of buds as observed above.

### 2.4. ssNMR Analysis of CAV1

Analysis of the membrane composition of lipid rafts and caveolae indicates that the most abundant phospholipids are ethanolamine- (PE) and choline-glycerophospholipids (PC). Moreover, the most abundant PE species in heterologous caveolae expressed in bacteria was 16:0-18:1 (palmitoyl-oleoyl) [43,44]. Consequently, the structural features of CAV1 were finally investigated when reconstituted into POPC/POPE/CHOL (molar ratio, 1:1:1) bilayer. A 2D ^13^C-^13^C ssNMR spectrum was then recorded to investigate the CAV1 morphology at the atomic level using the above systems. The CAV1 spectrum is typical for membrane proteins immobilized in lipid bilayers and allows for only the chemical shift assignment of several amino acids by type. In particular, the ^13^Cα-^13^Cβ cross-peaks corresponding to alanine (Ala), serine (Ser) and threonine (Thr) residues are easily recognized. The application of NMR chemical shifts in identification of protein secondary structures has a long history. Usually, it is accomplished by comparing the observed chemical shifts with the random coil values [45]. The averaged chemical shifts of 1HN, 15N, 1H, 13Cα, 13Cβ, and 13Cγ, together with the standard deviations categorized according to three secondary structure types, have been summarized according to the statistical data derived from a carefully prepared database containing >6100 amino acids in reference [46].

As shown in the spectrum (Figure 5A), almost all ^13^Cα/^13^Cβ Thr resonances highly overlap within a 3-ppm spectral region (65/68 ppm) that corresponds to α-helices. Perhaps only one Thr residue is in a random coil conformation as it gives rise to a small signal at ^13^Cα/^13^Cβ of 61.0/68.5 ppm. Since most of the Thr residues are found in the caveolin scaffolding domain (Figure 5B), this suggests α-helical structure for this domain. In addition, the chemical shift analysis indicates that most Ala residues are in α-helical conformation, some in random coil and a small fraction in β-sheet conformation. Similarly, the Ser residues are mostly within α-helices, with a significant fraction in random coils and none in β-sheets. Both Ala and Ser are evenly distributed in the primary sequence of CAV1, indicating that CAV1 is largely α-helical (including the scaffolding domain), partly random coiled, and very little β-sheet content, in agreement with CD results.

## 3. Discussion

Cell-free formation of caveolae-like structures have been accomplished in *E. coli*, Leishmania tarentolae (Lt) extract (LTE), and Protozoan Toxoplasma by inducing the heterologous expression of CAV1 [47,48,49]. As revealed by electron microscopy and functional assay, CAV1 generates membrane domains of similar size and curvature as the caveolae. However, the biological functions of the formed membrane curvature, such as endocytosis and binding with proposed protein partners, are compromised. These findings have revealed that although the stabilization and proper function of caveolae might depend on the existence of the specific protein partners, the correct folding of CAV1 and subsequent formation of caveolae-like membrane domains can be readily accomplished in the above systems. However, due to the complicated composition of the formed membrane domain, structural study of CAV1 in the above systems is hardly possible.

Here, we established a simplified model system using purified recombinant CAV1 and synthetic lipids. Recombinant caveolins have been previously prepared, however not necessarily with a focus on homogeneous, milligram-scale levels required for biophysical studies [50]. Alternatively, protein engineering was employed to increase caveolin solubility and bacterial expression levels at the expense of severely modifying the interaction of caveolin with membranes [30]. The massive generation of caveolae-like vesicles in *E. coli* by CAV1 over-expression strongly indicates that the protein properly folds when expressed in the bacteria and that post-translational modifications that normally occur in mammalian cells are not essential for the membrane scaffolding function of caveolins [51]. Thus, in the current work, the *E. coli* strain was selected for the expression of recombinant CAV1.

Obtained CAV1 was found to exist in the form of large oligomers, which can be broken down into monomers under very harsh conditions. The *N*- and *C*-terminus of CAV1 have been implicated previously in the CAV1 oligomerization and the interaction with CAV1 oligomers. In particular, the peptide CAV1(135-178) selectively interacts with itself and with the CAV1 *N*-terminus, but not with CAV2 or CAV3 [52]. This prompted us to investigate the self-aggregation behavior of the CAV1 *C*-terminus. Fibrillar structures formed by CAV1(168-178) provides a specific mechanism for CAV1 both to self-assemble on the membrane and to form linear structures that can further organize in a putative caveolin cage or scaffold. This also helps explain the difficulties in solubilizing CAV1 since amyloids are normally regarded as insoluble and irreversible.

CAV1 oligomers was reconstituted into lipid membrane using a mild detergent BOG, via the method of detergent dialysis, and lipid compositions holding different biological relevance were used. The CAV1 assemblies induced the formation of membrane bulges, while individual caveolae-like structure was not easily recognized. Clusters of CAV1 containing vesicles have also been observed in cytoplasm, before they were relocalized to the cellular membrane and formed surface constrained domain. However, mammalian cells can precisely modulate caveolin concentration and aggregation via protein expression and trafficking, and via interactions with other proteins, such as cavin1 and phospholipase Cβ1 [53,54]. In the current model, because of lack in control over the local protein concentration, the CAV1 aggregation is exacerbated during the generation of CAV1-liposomes via dialysis, which is independent of lipid species. Our findings provide more evidences for the statement that CAV-1 alone is enough to induce membrane curvature, as long as the local concentration of the protein is high enough. However, the size control and stabilization of the membrane curvature, as well as its proper involvement in various biological processes, such as endocytosis and signal transduction, depend on the existence of particular proteins and lipid species. Definition of caveolae-like structures formed by heterologous expression of CAV1 should be clearly distinguished from the caveolae in mammalian cells.

Despite its limited biological resemblance with the caveolae in vivo, the model system did provide a valuable platform for the structural study of CAV1. However, the resolution of ssNMR of ^13^C and ^15^N fully labeled CAV1 is not good enough for the specific assignment of each amino acid, which might be attributable to several reasons. These include low configuration of the ssNMR spetrometer, overlap of signals, large size of the samples, and so on. Multiple attempts will be made to optimize the sample in the future, in order to improve the resolution of ssNMR spectra, e.g., by protein engineering, screening liquid crystalline bilayers or employing selective isotope labeling [55,56,57].

One surprising finding is that the secondary structure of CAV1 scaffolding domain is determined to be α-helical by the chemical shift assignment of the amino acid Thr. Previous investigations of the caveolin scaffolding domain have shown that its secondary structure depends on the adjacent amino acids and lipids molecules, adopting either α-helix or β-sheet. Our finding demonstrated that when located in the membrane that mimics the caveolae, the scaffolding domain was α-helical within full-length CAV1, providing powerful evidences and clues for its further structural characterization. In addition, multiple evidences have shown that the scaffolding domain plays an important role in the interaction between CAV1 and various protein partners, and then mediates subsequent signal transduction. It is reported that CAV1 scaffolding domain may be critical in controlling the dynamic phenotype of cancer cells by facilitating the interaction of specific signal transduction pathways [58]. The peptide was able to competitively bind with various signal molecules and interrupt their interaction with CAV1 in vivo, exhibiting multiple biological activities [59,60,61]. The above findings suggest that the scaffolding domain might be a promising target for therapy of various diseases. Characterization of the structural features of the domain is of vital significance for the design and screening of bioactive compounds targeting CAV1.

## 4. Materials and Methods

### 4.1. Materials

DNA gene optimized for the bacterial expression of the 178 amino acids long CAV1 α-isoform was purchased from GenScript. This gene was cloned into the pET 31b (+) vector obtained from New England Biolabs. BL21(DE3) competent *E. coli* was purchased from New England Biolabs. Labeled ^13^C, and ^15^N M9 minimal media was prepared as previously described [62]. 2 × TY medium was prepared as following: 12 g tryptone, 7.5 g yeast extract, and 3.75 g NaCl were dissolved in 750 mL H_2_O. The medium was then sterilized by autoclaving and cooled down to room temperature before use. Isopropyl β-d-1-thiogalactopyranoside (IPTG) were purchased from Fisher Scientific. ^13^C-d-glucose and ^15^N-ammonium chloride were purchased from Sigma Aldrich. Sodium dodecyl sulfate (SDS) and Triton X-100 were purchased from Fisher Scientific. Empigen BB was purchased from Sigma Aldrich. BOG and LDAO were purchased from Anatrace. All the lipids, CHOL, and the extruder were purchased from Avanti Polar Lipids. Lipids were dissolved in chloroform and kept in 20 °C. The lipids used in current study include: DMPC, DPPC, POPC, POPE, and SM. Standard regenerated cellulose dialysis membrane tubing with cutoff of 1000 Da and 3500 Da, as well as tubing closures were purchased from Spectrum Labs. Whatman nuclepore track-etch membrane with pore sizes of 200 nm and 400 nm were purchased from Sigma Aldrich. 5 nm Ni-NTA-Nanogold for labelling of protein and NanoVan for negative staining were purchased from Nanoprobes.

Fluorenylmethyloxycarbonyl chloride (Fmoc) amino acids, 1-hydroxybenzotriazole monohydrate (HOBt), and *N*-[(1*H*-1,2,3-benzotriazol-1-yloxy(dimethylamino)methylene]-*N*-methyl-methanaminim hexafluorophosphate (HBTU), were obtained from CEM Corporation. Rink amide MBHA resin was obtained from EMD Millipore. Tris(hydroxymethyl)aminomethane (Tris), urea, imidazole, ethylenediaminetetraacetic acid (EDTA), acetonitrile, methanol, formic acid, piperidine, Diisopropylethylamine (DIEA), *N*,*N*-dimethylformamide (DMF), trifluoroacetic acid (TFA), 1-methyl-2-pyrrolidone (NMP), triisopropylsilane (TIS), 3,6-dioxa-1,8-octanedithiol (DODT), and isopropanol were obtained from Fischer Scientific.

HisPrepTM FF 16/10, a ready-to-use column, prepacked with precharged Ni Sepharose 6 Fast Flow was purchased from GE Healthcare. HiScreen Capto SP ImpRes and HiScreen Capto Q ImpRes columns (repacked with ion exchange chromatography medium) were purchased from GE Healthcare. Superdex 75 10/300 GL and Superdex 200 Increase 10/300 GL (prepacked gel filtration columns) were purchased from GE Healthcare. Jupiter 10u Proteo 90Å and Jupiter 10u C5 300Å were purchased from Phenomenex.

### 4.2. Plasmid Preparation

A DNA gene optimized for the bacterial expression of the 178 amino acids long CAV1 α-isoform was cloned into the pET 31b (+) vector between the NdeI and XhoI cloning sites, resulting in complete removal of the original gene for the ketosteroid isomerase protein and the addition of a small, *C*-terminal histidine tag (LEHHHHHH) to CAV1. The final vector was sequenced with standard T7 forward and reverse primers to confirm the cloning. The resulting vector was screened for expression using several *E. coli* strains and different conditions, including the cell density and temperature. CAV1 expression was found to be greatest in the BL21 (λDE3) strain with saturated cultures at 27 °C. *E. coli* BL21 (λDE3) transformed with this plasmid was plated and a dozen clones were tested for CAV1 over-expression by western blots of total cell SDS lysates. A clone that exhibited maximum protein expression was identified and a 15% glycerol stock of it was stored at −80 °C for subsequent protein expression.

### 4.3. Protein Expression

Cells were grown in 750 mL batches using 2.8 L baffled flasks in an Excella E24 incubator with shaking at 250 RPM. Cell growth was monitored by measuring apparent optical density at 600 nm (OD600) using a NanoDrop 2000 UV-VIS spectrophotometer (Thermo Fisher Scientific, Inc., Waltham, MA, USA). Ampicillin was used for bacterial selection at a concentration of 0.1 mg/mL. Fresh 2× TY medium was inoculated with over-night cultures at 37 °C and grown to an OD600 of approximately 3. For unlabeled CAV1, these cultures were cooled down to 27 °C for 30 min in the incubator. For uniformly [^13^C, ^15^N]-labeled CAV1, the cells were pelleted by 15 min of centrifugation at 5000× *g*, then suspended in M9 minimal medium supplemented with 2.5 g/L of ^13^C-glucose and 1 g/L of ^15^N-ammonium chloride, and allowed to accommodate to the new conditions for 30 min in the incubator at 27 °C. In both cases, the expression was then induced with 1 mM IPTG and allowed to take place at 27 °C for 4 h when the OD600 plateaued between 5 to 6. The cells were then harvested and stored at −20 °C for further protein purification.

### 4.4. Protein Purification

Frozen cells were thawed and chemically lysed by adding 0.1% Triton X-100, 0.8 M urea, 0.25 mg/mL lysozyme and Pierce universal nuclease, and incubated with stirring at room temperature for 1 h. Total membrane solubilization was achieved by the addition of 1.75% empigen BB detergent and further incubation at room temperature for 1 h. Insoluble material was removed by centrifugation at 7500 g and 4 °C for 1 h. Protein purification was accomplished using an AKTA Purifier 10 system (GE Healthcare, Pittsburgh, PA, USA) via a two-step strategy. Firstly, the total cell solubilized material was loaded onto a 20 mL HisPrep FF 16/10 column pre-equilibrated with 20 mM Tris, pH 8.0, containing 0.7% empigen BB, 100 mM NaCl and 60 mM imidazole. The column was then washed with 200 mL of the same buffer and then with 50 mL of the above buffer with 1 M urea and 60 mM imidazole. Secondly, CAV1 eluted from the Ni resin using the above buffer with 1 M urea, and 0.5 M imidazole, was then directly applied onto a 4.7 mL HiScreen Capto Q anion exchange column. Detergent exchange from empigen BB to BOG was accomplished by washing the column with 50 mL of 20 mM Tris, pH 8.0, containing 60 mM BOG. Finally, CAV1 was eluted with a linear gradient of NaCl from 0.2 to 0.4 M. Purified protein was stored in 20% glycerol at −20 °C.

### 4.5. Protein Characterization

Purified CAV1 was analyzed by SDS-PAGE using Coomassie blue staining and by western blotting using an anti-pentahistidine antibody conjugated with horseradish peroxidase. Samples were boiled at 95 °C for 10 min before loading. Separation of monomeric, unfolded CAV1 was accomplished firstly by precipitation of the BOG-solubilized protein with methanol/chloroform and re-solubilization in neat formic acid. The sample was then diluted with FMA (20% each of formic acid, methanol and acetonitrile in water) and separated by size-exclusion chromatography on a Sperdex 75 column in FMA at the flow rate of 1 mL/min. The fraction corresponding to monomeric CAV1 was collected and used immediately. This sample was used to determine (1) the CAV1 molecular weight by direct injection into an electrospray ionization quadrupole time-of-flight mass spectrometer (EI-MS) (Waters, MA, USA) and (2) the CAV1 *N*-terminal amino acid sequence by lyophilization and subsequent Edman degradation on a Procise 494 protein sequencer (Applied Biosystems, Lakewood, CA, USA).

### 4.6. In-Gel Tryptic Digestion

Specifically, CAV1 was analyzed with SDS-PAGE as described above firstly and the gel was stained with Coomassie brilliant blue R-250 as usual. The bands corresponding to monomeric CAV1 was then excised using a scalpel and cut into small pieces. 500 μL of destaining solution (2 mg/mL ammonium bicarbonate in 50% acetonitrile) was added to the gel pieces and the mixture was incubated for 30 min with shaking at 37 °C. Destaining was repeated for several times until all the dye was removed. 50 µL of acetonitrile was added to shrink the gel pieces and carefully removed by air dry. Swelling of gel pieces was accomplished by adding 10 μL of activated trypsin solution (0.01 μg/μL trypsin and 2 mg/mL ammonium bicarbonate) to the tube and further incubation at room temperature for 15 min. To fully digest the protein, more digestion buffer (2 mg/mL ammonium bicarbonate) was added and the mixture was incubated at 30 °C overnight with shaking. The digestion mixture was recovered by centrifugation at 10,000× *g* and ready for further liquid chromatographic separation and EI-MS.

### 4.7. Membrane Reconstitution

CAV1 purified in BOG was reconstituted into a membrane system of several phospholipid composition: DMPC/CHOL, DPPC/CHOL, and DPPC/SM/CHOL by detergent dialysis. Lipids and CHOL were dissolved in chloroform and stored at −20 °C. Required aliquots were mixed and the chloroform was then removed under low vacuum followed by overnight lyophilization. The dried lipid mixture was then dissolved in 20 mM Tris, pH 8.0, containing 60 mM BOG at a final concentration of 2.5 mg/mL and vortexed until clear. BOG-solubilized lipids and CAV1 were then mixed and incubated at 30 °C for 1 h. After the incubation, the mixture was dialyzed using a 3.5 kDa cutoff membrane against 50 mM NaCl, 2 mM EDTA, 20 mM Tris, and 0.1 M urea for 24 h. The liposomes were recovered via ultracentrifugation at 180,000× *g* for 4 h.

### 4.8. TEM

Preformed liposomes were dialyzed against 20 mM Tris, 50 mM NaCl and 8 mM imidazole, pH 8.0 for 4 h and then incubated with 5 nm Ni-NTA-Nanogold (Nanoprobes) for 30 min at room temperature. 10 μL of liposome/nanogold mixture was placed on thin carbon films on holey grids and allowed to absorb for 2 min, after which the grid was washed with 10 μL of deionized water twice and negatively stained with methylamine vanadate. Imaging was carried out with a Tecnai G2 transmission electron microscope (FEI, OR, USA) operated at 80 kV.

### 4.9. CD Spectroscopy

Since a CD signal will be observed only when a chromophore is chiral (optically active), most of the lipid molecules do not produce CD signals. CD signals of proteins and peptides arise from peptide bonds (far UV region in the range 190 to 250 nm, the information of the secondary structure), the aromatic amino acid side chains (near UV region in the range of 250 to 290 nm, about tertiary folding) and the disulfide bonds. The conformation of CAV1 in 60 mM BOG and in the lipid bilayer was obtained using a Model 202SF CD spectrometer (Aviv Associates, Lakewood, NJ, USA). Spectra were obtained from 250 to 190 nm in 1 nm increments at 25 °C and the reported spectra correspond to the average of at least three wavelength scans. The CD spectra of proteins reconstituted in liposomes was acquired by subtracting the CD signal of liposomes from that of proteoliposomes.

### 4.10. Peptide Synthesis and Purification

The peptide with the sequence of SNVRINLQKEI corresponding to CAV1(168-178) was synthesized by standard Fmoc-based solid phase peptide synthesis on a microwave-based Liberty instrument (CEM, Stallings, NC, USA) and was capped with a *N*-terminal *N*-methyl amide group by employing a rink amide MBHA resin and with a *C*-terminal acetyl group. Factory-default single coupling steps were applied for each amino acid except for arginine (double coupling); deprotection was performed in piperidine/HOBt and coupling occurred in HBTU/DIEA. Solvents used for washing steps were DMF and NMP. Cleavage of the peptide from the resin and removal of side-chain protecting groups were accomplished by incubating the resin for about two hours with 92.5% TFA, 2.5% DODT, 2.5% TIS and 2.5% water at room temperature. The peptides were precipitated twice with cold ether and dried under low vacuum followed by high vacuum overnight.

The peptide was purified by HPLC on a C12 column (Phenomenex) on a AKTA Purifier 10 (GE Healthcare, Pittsburgh, PA, USA). Peptides were initially dissolved in neat formic acid assisted by sonication and then diluted with acetonitrile to a final clear solution with 10% formic acid and 30% acetonitrile. The solvent system employed for the purification was: (A) 0.1% TFA, 10% acetonitrile and (B) 0.1% TFA, 90% acetonitrile. The fractions containing the desired peptide were collected manually and immediately dried by lyophilization and subsequently stored at −20 °C. A small amount of purified peptide was dissolved in formic acid, diluted with acetonitrile and directly injected on an EI-MS to confirm its identity and purity. The following *m/z* values were obtained: 1354.6 ([M]^+^) and 677.7 ([M]^2+^) corresponding to a molecular weight of 1354.6 Da (expected 1354.6 Da).

### 4.11. ThT Binding Assay

The synthesized CAV1(168-178) peptide was dissolved in DMSO and then diluted with 20 mM Tris, pH 8.0 to 1 mg/mL. The peptide solution was incubated at 37 °C for 48 h before ThT assay and TEM imaging. CAV1 in 20 mM Tris, pH 8.0, containing 60 mM BOG at a concentration of about 1 mg/mL was incubated at room temperature for one day and then used for ThT assay and TEM imaging. 100 μL of 50 μM ThT was added into 5, 10, and 15 μL aliquots of CAV1(168-178), CAV1 and the corresponding buffer (as controls), respectively. The fluorescence was measured with a SpectraMax M2 Multimode Plate Readers (Molecular Devices, San Jose, CA, USA) with excitation at 446 nm and emission at 485 nm.

### 4.12. ssNMR Spectroscopy

Uniformly [^13^C, ^15^N]-labeled CAV1 was reconstituted into POPC/POPE/CHOL (molar ratio of 1:1:1), mimicking the lipid composition of caveolae, as described above. The liposomes were harvested by ultracentrifugation at 175,000 g for 2 h at 4 °C. The pellet was then deposited on clean glass, allowed to dry under controlled humidity, packed into 3.2 mm MAS rotors and rehydrated with water. The experiments were carried on a 400 MHz NMR spectrometer using a ^1^H/^13^C/^15^N 3.2 mm magic angle spinning (MAS) probe (Bruker, Bremen, Germany). 100 kHz ^1^H decoupling was employed and 50 kHz ^13^C and ^15^N fields were used for 90-degree pulses and during cross polarization blocks. In order to eliminate the natural abundance ^13^C signal of lipids, double-quantum spectroscopy was employed via a 0.2 ms HORROR sequence on a spectrometer with the ^1^H frequency of 400 MHz, at 20 kHz MAS and −10 °C (the actual sample temperature being around 10 °C) [63].

## Figures and Tables

**Figure 1 molecules-26-06201-f001:**
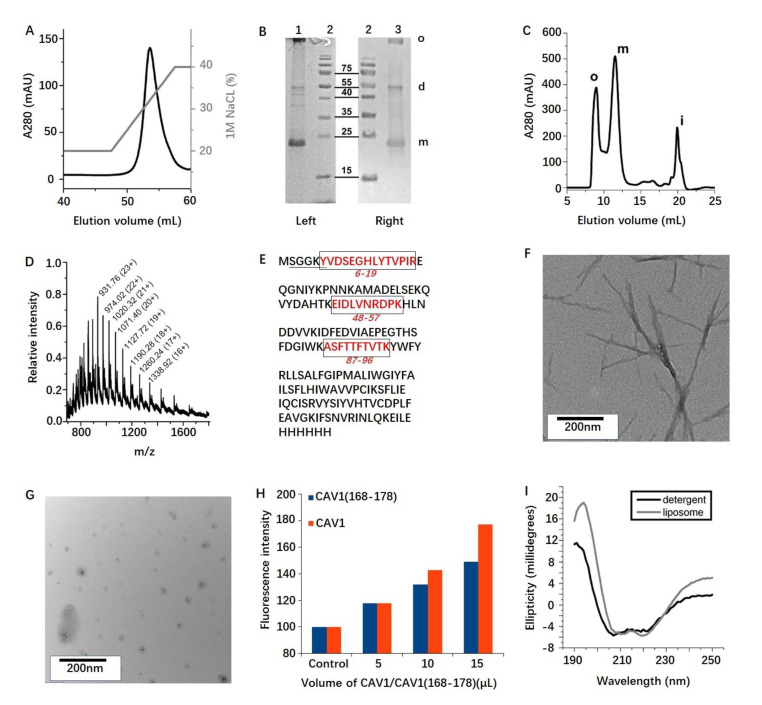
Molecular characterization of CAV1. (**A**) CAV1 elution profile (black, left axis) during ion exchange chromatography using a gradient of 1 M NaCl (gray, right axis). (**B**) Coomassie-stained SDS-PAGE gel (**left**) and Western blot (**right**) analysis of purified CAV1 (lanes 1, 3). The marker is shown in lane 2 with several molecular weights in kDa. The bands in lane 3 refer to monomeric (m), dimeric (d) and oligomeric (o) CAV1, respectively. (**C**) CAV1 purification by gel filtration in FMA solvent: oligomeric (o) and monomeric (m) CAV1, and residual impurities (i). (**D**) Mass spectrum of monomeric CAV1 as purified in (**C**). (**E**) *N*-terminus amino acids determined by Edman degradation on a Procise 494 protein sequencer (underlined) and short peptides obtained by in gel trypsin digestion (red). (**F**) Transmission electron microscopy (TEM) of negatively stained amyloid fibrils formed by CAV1(168ߝ178) peptide. (**G**) TEM of negatively stained aggregates formed by CAV1. (**H**) Thioflavin-T binding assay. Control (20mM Tris, pH 8.0 for CAV1(168–178); 20mM Tris, pH 8.0, 60 mM BOG for CAV1), 1 mg/mL of either aggregated CAV1(168–178) (blue) or BOG-solubilized CAV1 (red) were incubated with 100 μL of 50 μM ThT. The fluorescence was measured and plotted against the control. (**I**) CD of CAV1 when solubilized in BOG detergent (black) and reconstituted in 1,2-ditetradecanoyl-*sn*-glycero-3-phosphocholine (DMPC)/Cholesterol (CHOL) membranes (grey).

**Figure 2 molecules-26-06201-f002:**
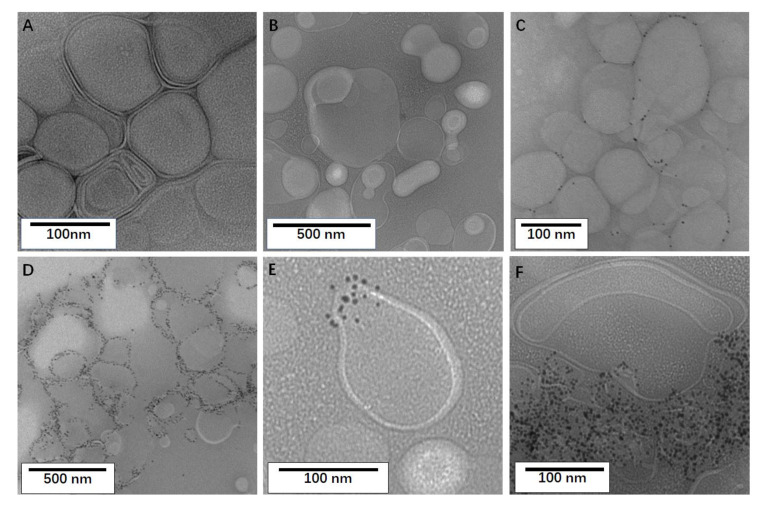
Representative TEM images of DMPC/CHOL (molar ratio, 1:1) liposomes without protein (**A**,**B**), with CidA (**C**), LrgA (**D**) and CAV1 (**E**,**F**). The protein-to-lipid weight ratios were 1:30 in C, 1:5 in D and 1:20 in E and F.

**Figure 3 molecules-26-06201-f003:**
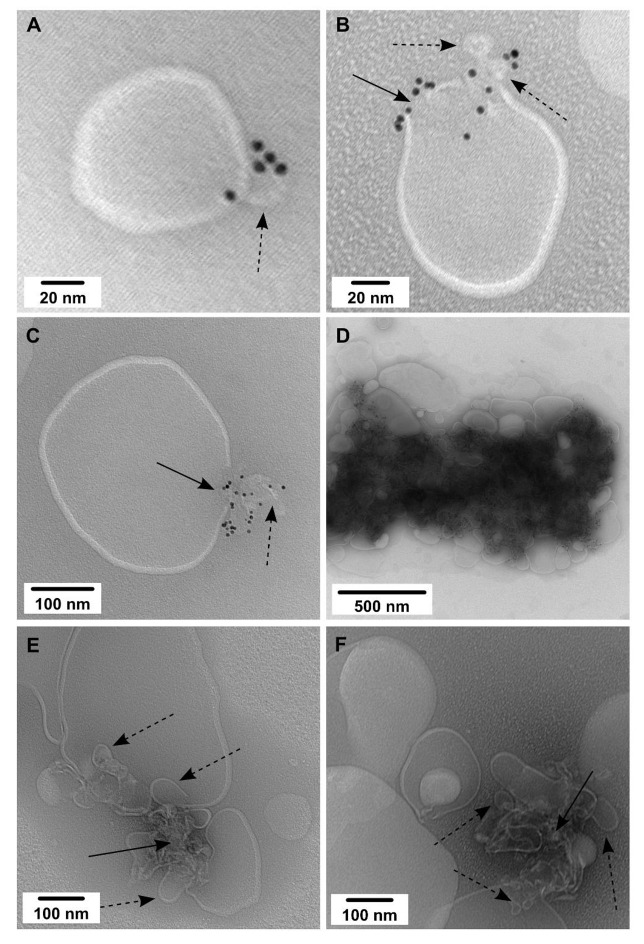
TEM images of CAV1 reconstituted in DPPC/CHOL (molar ratio, 1:1) liposomes. The protein-to-lipid weight ratios was 1:20. Small (**A**,**B**) and large (**C**) CAV1-induced buds are observed intact or ruptured (labeled by dotted arrows); solid arrows indicate membrane discontinuities at ruptured buds. CAV1-membrane aggregates labeled (**D**) and unlabeled (**E**,**F**) with Ni-NTA nanogold. In (**E**,**F**) well-defined buds are observed (labeled by dotted arrows) around very dense areas (labeled by solid arrows).

**Figure 4 molecules-26-06201-f004:**
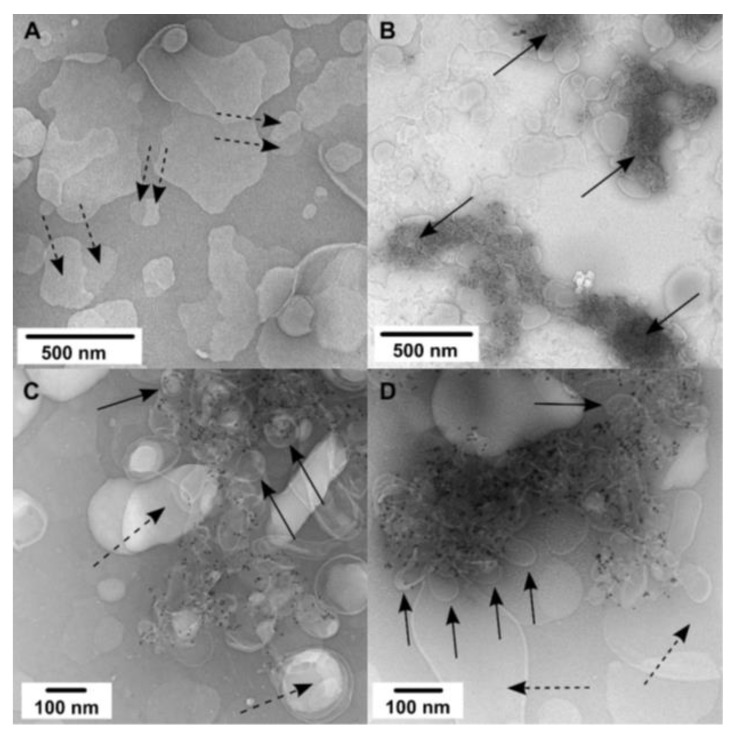
DPPC/SM/CHOL (molar ratio, 3:1:4) liposomes without (**A**) and with CAV1 (**B**–**D**). The protein-to-lipid weight ratios was 1:20. Dotted arrows point to protein-free membranes and highlight apparent lipid domains in (**A**). Solid arrows indicate CAV1-induced buds: either as large clusters (**B**) or well-defined ones (**C**,**D**).

**Figure 5 molecules-26-06201-f005:**
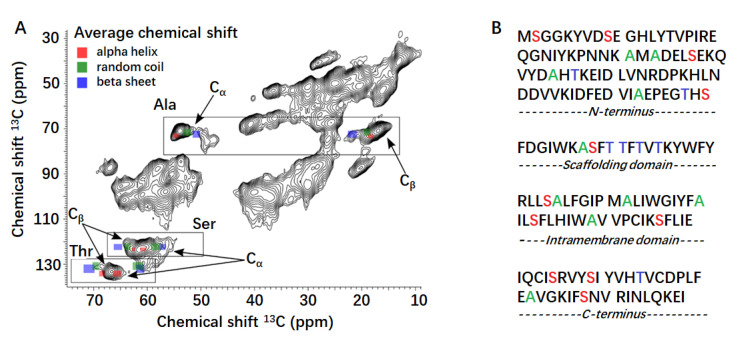
(**A**) 2D ssNMR spectrum of uniformly [^13^C, ^15^N]-labeled CAV1 reconstituted in 1-hexadecanoyl-2-(9*Z*-octadecenoyl)-*sn*-glycero-3-phosphocholine (POPC)/ 1-hexadecanoyl-2-(9*Z*-octadecenoyl)-*sn*-glycero-3-phosphoethanolamine (POPE)/CHOL. The protein-to-lipid weight ratios was 1:5. The ^13^Cα-^13^Cβ cross-peaks for Ala, Ser and Thr residues are highlighted by enclosing rectangles. The typical secondary structure-dependent chemical shifts for ^13^Cα and ^13^Cβ (both the average and the standard deviation values as taken from [46]) are depicted by red (α-helix), green (random coil) and blue (β-sheet) rectangles centered at the corresponding average value and having the width/height equal to the standard deviation. (**B**) Distribution of Ala (green), Ser (red) and Thr (blue) residues in the primary sequence of CAV1. Different CAV1 domains are presented as labeled above.

## Data Availability

The data presented in this study are available in the present article.

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
