# Peer review of "Reconstitution of Caveolin-1 into Artificial Lipid Membrane: Characterization by Transmission Electron Microscopy and Solid-State Nuclear Magnetic Resonance"

_molecules, 2021, doi:10.3390/molecules26206201_

Round 1
Reviewer 1 Report
The paper reports results about the study of the membrane protein Caveolin1. The results reported are interesting and contribute to the knowledge about this protein that can be considered as a terapeutic target for many diseases.
The paper thus is to be published. To my opinion only some modifications may ameliorate the text. In the introduction the sentence from row 56 to 84 should be divided into brief sentences to help the less familial reader to the previously reported results about this protein nwith attentio to the shorter forms and the aggregation states.
In Fig. 1 the comparison between CDs should be reported as usual in mean molar ellipticity per residue. Probably the two curves ( BOG and DMPC) may be parallel to these wirh different values but this does not decrease the validity of the results reported.
The spectra 13C reported and where the identification of the helical tracts are located in the structure have been obtained from caveolin1 labelled in 13C but the details of the procedure adopted to apply the algorithm of Sykes et al. are not reported . Prticularly in the roews 281-290 should be implemented to help the reader to follow the structural details obtained. Moreover comment about the addition in this experiment of phosphatidylethanolamine ( negatively charged) to the previously used phosphatidylcholine-CHO ( dipolar) should be added to the text.
The utility of the 15N labelling is not reported. Which is the utility if this labelling?.
Reviewer 2 Report
In this study the authors report on the expression and purification of recombinant CAV1, a membrane protein involved in the formation of specialized cellular membrane domains, the caveolae, and its C-terminal peptide. The oligomerization state of the protein is discussed. The protein is also reconstituted into artificial membranes made of various synthetic lipids. The effect of the protein on the morphology of the membrane is studied by TEM and structural information is also obtained by solid state NMR and circular dichroism.
Overall this study is interesting and it shows that CAV-1 alone can induce the curvature of the membrane. Therefore, i believe that this article is suitable for publication in Molecules after the following minor corrections:
1) Line 46: Are CAV 1, 2 and 3 really isoforms (ie similar proteins originating from a single gene or gene family) What is confusing for me is that later in the manuscript (line 57) the authors mentions the existence of two isorforms of CAV 1.
2) line 47 : "formation of caveloae formation" One of the word formation should be removed.
3) line 72: "are 8S in size" . I believe that S is not correct.
